# Melatonin: A Neurotrophic Factor?

**DOI:** 10.3390/molecules27227742

**Published:** 2022-11-10

**Authors:** Armida Miranda-Riestra, Rosa Estrada-Reyes, Erandis D. Torres-Sanchez, Silvia Carreño-García, Genaro Gabriel Ortiz, Gloria Benítez-King

**Affiliations:** 1Laboratorio de Neurofarmacología, Subdirección de Investigaciones Clínicas, Instituto Nacional de Psiquiatría Ramón de la Fuente Muñiz, Calzada México-Xochimilco 101, San Lorenzo Huipulco, Tlalpan 14370, Mexico City, Mexico; 2Laboratorio de Fitofarmacología, Dirección de Investigaciones en Neurociencias, Instituto Nacional de Psiquiatría Ramón de la Fuente Muñiz, Calzada México-Xochimilco 101, San Lorenzo Huipulco, Tlalpan 14370, Mexico City, Mexico; 3Departamento de Ciencias Médicas y de la Vida, Centro Universitario de la Ciénega, Universidad de Guadalajara, Ocotlán 47810, Jalisco, Mexico; 4Dirección de Investigaciones Epidemiológicas y Psicosociales, Instituto Nacional de Psiquiatría Ramón de la Fuente Muñiz, Calzada México-Xochimilco 101, San Lorenzo Huipulco, Tlalpan 14370, Mexico City, Mexico; 5Departamento de Disciplinas Filosóficas y Metodológicas, Centro Universitario de Ciencias de la Salud, Universidad de Guadalajara, Guadalajara 44340, Jalisco, Mexico

**Keywords:** melatonin, neurogenesis, pleiotropic, neurotrophic factors, neural differentiation

## Abstract

Melatonin, *N*-acetyl-5-hydroxytryptamine, is a hormone that synchronizes the internal environment with the photoperiod. It is synthesized in the pineal gland and greatly depends on the endogenous circadian clock located in the suprachiasmatic nucleus and the retina’s exposure to different light intensities. Among its most studied functions are the regulation of the waking-sleep rhythm and body temperature. Furthermore, melatonin has pleiotropic actions, which affect, for instance, the modulation of the immune and the cardiovascular systems, as well as the neuroprotection achieved by scavenging free radicals. Recent research has supported that melatonin contributes to neuronal survival, proliferation, and differentiation, such as dendritogenesis and axogenesis, and its processes are similar to those caused by Nerve Growth Factor, Brain-Derived Neurotrophic Factor, Neurotrophin-3, and Neurotrophin-4/5. Furthermore, this indolamine has apoptotic and anti-inflammatory actions in specific brain regions akin to those exerted by neurotrophic factors. This review presents evidence suggesting melatonin’s role as a neurotrophic factor, describes the signaling pathways involved in these processes, and, lastly, highlights the therapeutic implications involved.

## 1. Introduction

Melatonin (MEL) *N*-acetyl-5-methoxytryptamine is a hormone synthesized by the pineal gland. Discovered and isolated from bovine pineal glands by Aaron Lerner, MEL was injected in Lerner’s groundbreaking experiment into dermatosis patients who, contrary to his expectations, did not experience depigmentation but drowsiness. With these initial findings, he began to investigate MEL [1]. Later, MEL was described in plants as phytomelatonin [2,3,4], and, nowadays, it is known that MEL is present in practically all organisms along the phylogenetic scale [5].

The pineal gland produces MEL in the dark phase of the photoperiod. MEL is secreted into the general circulation and in the cerebrospinal fluid and, in this manner, circulates throughout the body and the brain [6]. Its synthesis is regulated by a clock located in the suprachiasmatic nuclei (SCN) at the hypothalamus. This clock determines the circadian rhythm of MEL secretion. In the brain, extrapineal MEL behaves similarly to Neurotrophic Factors (NTFs) [7,8] and is capable of modulating cell survival, proliferation, and differentiation, by signaling pathways that can be triggered in response to stimulation of membrane and intracellular receptors.

In this regard, NTFs play a crucial role in brain neuroplasticity and neurodevelopment, promoting its growth and survival [9,10]. These peptide molecules function as signals that trigger biological processes allowing adaptation to the environment and survival of neurons. In the early stages of development, they modulate both differentiation and maturation of neuronal precursors. Some factors are only present at the early stages of development and others throughout life [10,11,12]. Besides neurogenesis, NTFs are also needed for the maintenance of neuronal function and the neuron’s structural integrity [11,13]. NTFs are also expressed in non-neuronal tissues like lung components including nasal and bronchial epithelium, smooth muscle, nerves, immune cells, kidney, spermatozoa, and ovarium [9,14,15]. These peptides are secreted into the cellular medium and may act in a paracrine manner signaling neighboring cells. If required, cells can also synthesize and secrete these factors in response to autocrine stimulation [16].

MEL can also be synthesized in other organs such as the intestine, retina, placenta, specific brain regions, and skin [17,18] (Figure 1). MEL is released at micromolar concentrations from extrapineal sites of synthesis and has paracrine, autocrine, and antioxidant actions [19]. The pleiotropic effects of indolamine have been widely described. For this reason, in this review, we will compare NTFs’ features with MEL synthesis, focusing on their role in neurogenesis in the adult brain, as well as on the mechanisms of action involved in survival, proliferation, and neurodevelopment. 

## 2. Melatonin and Neurotrophic Factors Synthesis

MEL is an indolamine with a 3-amido group and a 5-methoxy group, which confers its amphiphilic properties to MEL (*N*-[2-(5-methoxy-1*H*-indol-3-yl)ethyl] ethanamide) [20]. MEL biosynthesis starts from the amino acid tryptophan which is transformed into serotonin. In the pineal gland, serotonin will undergo two enzymatic reactions: First, *N*-acetylation by *N*-acetytransferase (NAT) will produce *N*-acetyl-serotonin. Then, a methyl group is transferred from *S*-adenosylmethionine to the 5-hydroxy group of *N*-acetylserotonin by the action of hydroxyindol-*O*-methyltransferase (HIOMT) [21]. NAT is the limiting step for MEL synthesis, and it has a high amplitude rhythm with a nocturnal activity that is 50 to 100 times greater than diurnal activity. HIOMT, on the other hand, catalyzes the last step of the synthesis and has a very low rhythmic amplitude (Figure 2) [21].

MEL in circulation is primarily catabolized in the liver and the brain. In the hepatocytes, MEL undergoes 80% of hydroxylation by the cytochrome p450 1A2 (CYP1A2); afterward, in the kidney, it is sulpho-conjugated (70 to 80% of total catabolism) or conjugated with glucuronic acid (5% of total catabolism) [22]. CYP1A2 ensures hepatic catabolism [23]. Even at the hepatic level, there are also deacetylation and demethylation pathways. After deacetylation, MEL is transformed into 5-methoxytryptamine and after oxidation to 5-methoxyindole acetic acid [24]. Cleavage of the pyrrole nucleus of MEL results in *N*-acetyl-5-methoxykynurenamine through the *N*-formyl intermediate [24]. 70% of this metabolite is eliminated by urine and 20% in the feces in 24 h. The main route of elimination is, therefore, urinary and will lead to the formation of various metabolites present in the urine: 6-hydroxymelatonin in the form of sulfate (70–80%) and glucuronide (5%) and kynuramines (15%), among them. In summary, MEL urinary elimination comprises MEL (1%), 5-methoxyindoleacetic acid (0.5%), and 6-sulfatoxymelatonin, which is the major urinary metabolite [25].

On the other hand, NTFs have significant differences in their biosynthesis and catabolism. MEL is an indolamine, while NTFs are peptides synthesized from protein precursors. Despite the great number of molecules considered as NTFs that can regulate neuronal function, we will focus on just four of them with similar structure and functions: Nerve Growth Factor (NGF), Brain-Derived Neurotrophic Factor (BDNF), Neurotrophin-3 (NT3), and Neurotrophin-4/5 (NT4/5).

NTFs are synthesized by neuronal cells as inactive precursors (pro-NTFs, 27 kDa). Synthesis of Pro-NTFs occurs in the endoplasmic reticulum and then a cleavage is caused by furin enzymes in the intracellular compartment to produce active molecules. Also, cleavage is produced by plasmin metalloproteinases in the extracellular space. Pro-NTFs are further processed to generate mature NTFs nearly 13–15 kDa polypeptides with extensive homology. The mature forms of BDNF, NT4, and NT3 have approximately 50% amino acid identity to NGF [9,26]. Mature NTFs form homodimers by noncovalent binding and contain a conserved terminal fold and a cysteine “knot”, which consists of three disulfide bonds in the polypeptide chain [16,27]. 

## 3. The Neurodevelopment in the Adult Brain

The organisms could survive and adapt to their environment through neuroplastic changes produced in the brain. Neuronal activity can modify the number and strength of synaptic connections [28]. The activity-dependent modulation of synapses is a critical factor for brain development alongside many cognitive functions and behavior. It has been postulated that from the embryonic stages and during adulthood, various cytotypes, tissues, and organs release NTFs. These factors are essential for new neuron formation, their survival as well as differentiation, and the correct organization of the nervous system [29].

NTFs expression is regulated by neuronal activity and these molecules modulate the efficiency of synaptic transmission, dendrite, and axon growth, as well as the elements necessary for synaptogenesis [30]. These processes are adaptative changes in which the number of neurons correlates with needs and the number of neurons in the innervated target neurons. The “*neurotrophic factor hypothesis*” was conceptualized by Victor Hamburger and Rita Levi Montalcini, who identified the first of these factors in NGF [31]. In the adult mammalian brain, there are several neurogenic niches: the hippocampal dentate gyrus, the subventricular zone of the lateral cerebral ventricle, the substantia nigra and the cerebellum, the amygdala, the spinal cord, the hypothalamus, and the cerebral neocortex [32,33].

Neurogenesis and neuronal differentiation take place at three stages: (1) Nervous cell generation through cellular proliferation, (2) Migration of neuronal precursor to the area where they will be established, (3) Morphofunctional polarization of neurons in the somatodendritic and axonal domains [34]. Communication between distal neurons occurs in the third stage of neurogenesis, in which axons and dendrites are formed, synapses are formed according to electric activity patterns, and nonfunctional connections are eliminated [34]. This process begins with the neuronal secretion of neurotrophic factors. The nerve terminals of neighbor neurons capture these factors, internalize them, and transport them in a retrograde direction to the soma. Growth factors are constantly secreted to establish functional connections, as shown in Figure 3. Finally, the neurons that are not exposed to the secreted NTFs enter apoptosis and die (Figure 3).

In a similar fashion as NTFs, pineal MEL and extrapineal MEL, which are released at specific brain regions, promote and improve neurogenesis, synaptogenesis, and growth of axons as well as dendrites. This claim will be addressed in the next paragraphs by describing the evidence about MEL effects on these neurodevelopmental processes and by answering our guiding question: Does MEL act on neuronal plasticity and the formation of new synapsis as the NTFs, even though it is a low mass indole molecule?

## 4. Evidence That Supports Melatonin Acts as a Neurotrophic Factor

Like the neurotrophic factors, MEL also participates in brain neuroplasticity and neurodevelopment [37,38]. One important feature of neurogenesis is the proliferation and survival of neuronal precursors. MEL in vitro at 10^−6^ M stimulates both processes [39]. Moreover, with MEL at 10^−7^ M, the formation of new neurons increased by approximately 70% in comparison to the control. On the other hand, the indolamine administered for 14 days to mice increases the number of “newborn” neurons in the dentate gyrus of the hippocampus [40]. However, there was no increase in the proliferation rate. Instead, an increase in the survival of neuronal precursors was observed during the proliferation process. By contrast, other authors observed increased proliferation of new neurons (158%) in the subventricular zone of the adult mouse brain and rat embryos, with concentrations ranging between 10^−7^ M and 10^−6^ M [39,41]. These effects were dose-dependent because higher MEL concentrations increased cell survival [39,41]. Environmental factors in combination with MEL also promote neuroplastic changes, for instance, physical activity plus MEL administration enhances neurogenesis in mice [42].

Recently, we found increased neurogenesis in human olfactory neuronal precursors incubated with MEL, ketamine, and a combination of these compounds. This effect was similar to what has been observed in relation to neurotrophic factors BDNF, Epithelial Grow Factor (EGF), and Fibroblast Growth Factor (FGF) in the dentate gyrus of the hippocampus. The increased neurogenesis was associated with the antidepressant-like behavior produced by these NTFs and the combination MEL/KET [43,44].

MEL also stimulates dendritogenesis, dendrite’s spine formation, dendritic arborization, and synaptogenesis [37,45]. The indolamine maintains the neuronal somatodendritic domain [46] through structural morphofunctional polarization thanks to cytoskeletal rearrangements. The three cytoskeletal components (microtubules, microfilaments, and intermediate filaments) are reorganized in the presence of this indolamine [46]. Hence, an important function of MEL for neurodevelopment is the modulation of the cytoskeletal organization. Cytoskeletal rearrangements play a crucial role in the formation and enlargement of axons and dendrites, as well as in the synaptic assembly (for review see [46,47]).

Regarding dendritogenesis, there is a vast literature about the effects of MEL in dendrite formation in organotypic cultures of hippocampus and animal models. Besides the survival-promoting effect, MEL also increases new neuron maturation in adult brains [48], by augmenting the ramifications of the dendritic trees in the hilus of the hippocampus. In this sense, the systemic administration of MEL to mice for 14 days produced an increase in the number of dendritic arborizations. Of utmost importance is that the loss of dendrites in epilepsy and Alzheimer’s (AD) diseases also occurs in this brain region [45,49] and that enhanced dendritic complexity, which is measured in arborizations in the dorsal-ventral regions of the dentate gyrus in male Balb/C mice, is associated with an antidepressant-like behavior [50]. MEL has also a neuroprotective effect on dendrite formation, for example, administration of MEL to rats, which were previously submitted to global ischemia, prevents the impairment of place learning and memory, both of which are integrated into the hippocampus. Just as importantly, MEL partly preserves the density of spines, mushroom spines, and dendrites pyramidal neurons, which are necessary for an adequate synapsis formation [51]. The indolamine not only counteracted and protected the neurons against apoptosis; it also prevented and reversed the dendritic arborization retraction [52], and fostered synapse formation in hippocampal organotypic cultures at 100 nM. This was evidenced by staining with an anti-synapsin antibody which labels synapsin, a protein localized in the presynapsis [45].

The evidence that supports the stimulation of axogenesis by MEL was obtained in cultured neurons and brain tissue. In cultured human olfactory neuronal precursors, MEL at 10^−7^ and 10^−5^ M increased by 15% the axonal formation [37]. Moreover, neurite formation—the primary neurodevelopment step that antecedes axogenesis—is stimulated by MEL in N1E-115 cells [47]. In addition, experiments in rodents support that MEL stimulates axogenesis. Daily administration of MEL at 8 mg/kg for one or six months to male Balb/C mice increased the granular cell layer in the dentate gyrus by 11–33% and the volume of suprapyramidal and infrapyramidal mossy fiber projection of granule neurons in the dentate gyrus of the hippocampus. Also, an increase in the volume of the CA3 region was observed in this work [53]. Recently, we observed spine formation in a clone of human olfactory neuronal precursors stimulated by MEL (preliminary data). The evidence described in this section suggests that MEL, analogous to NTFs, promotes the distinct stages of the neurodevelopmental process in the adult brain. Therefore, it is possible to consider that MEL acts as a neurotrophic factor. Although, in order to highlight its similarities to NTFs, it is relevant to present the pathways underlying neurogenesis, proliferation, and neuronal survival, pathways that reveal the role of MEL in these stages [44].

### Mechanism of Action Involved in Neurogenesis and Neural Differentiation: Neurotrophic Factors and Melatonin

The biological activity of NTFs and MEL occurs thanks to their interaction with transmembrane receptor proteins. In the case of NTFs receptors, they have an extracellular domain, where the NTFs bind [12,26], and a cytosolic domain with catalytic and regulatory activity. In contrast, MEL receptors have seven transmembrane domains with an extracellular N-terminal and an intracellular C-terminal [54]. Given that the transactivation of Trk receptors is well documented [55,56], we will briefly describe the pathways involved in neurogenesis, proliferation, and survival. MEL participates in all these stages (Figure 4).

There are two types of receptors for NTFs: a low NTFs affinity pan-NTF receptor p75NTR that belongs to Tumoral Necrosis Factor (TNF) family, and the high-affinity tropomyosin-related kinase (Trk) receptors; both activate multiple signaling pathways [26,29,56,57]. NGF preferentially activates TrkA, BDNF, and NT-4 activates TrkB, while NT-3 activates TrkC [58]. Once NTFs are bound, the receptor dimerizes and phosphorylates itself on the cytoplasmic domain. When the receptor is phosphorylated, it forms the core of the binding site, adapting proteins and enzymes that mediate the rapid (seconds to minutes) activation of downstream signaling cascades. The main signaling pathways activated by Trk receptors are Ras, phosphoinositide-3 kinase (PI3K), and phospholipase (PLC-γ1); which in turn activate their downstream effectors [30], including stimulation of the mitogenic protein kinase (MAP) cascade and protein kinase B (Akt). Afterward, diacylglycerol (DAG) and inositol triphosphate (IP3) are produced by IP3K activity and PLCγ1, leading to calcium (Ca^2+^) mobilization. These pathways activate transcription factors involved in cell differentiation, survival, growth, and apoptosis, all of which are processes that occur in hours or days. In addition, NTFs modulate plasma membrane receptors, such as NMDA which are ionic channels permeable to Ca^2+^ and Na^+^; both Ca^2+^ and Na^+^ are crucial for neuronal function and differentiation [59,60].

Melatonin receptor 1 (MT_1_) and melatonin receptor 2 (MT_2_) share amino acid sequences. However, other sequences distinguish them as “fingerprints”. MT_1_ and MT_2_ receptors are coupled to Galpha proteins which are stimulatory (Gs) or inhibitory (Gi). MT_1_ and MT_2_ can be coupled to Gi which in turn inhibits the cAMP production or to a Gq which activates beta-type PLC-β leading to the production of PIP2, IP3, and DAG [54,61]. Similarly, to NTFs that activate TrKs receptors, MEL activates the same signaling pathways but through its binding with its specific receptors. In addition, MEL can cross the plasmatic membranes thanks to its amphiphilic features and binds to intracellular proteins such as calmodulin (CaM), calreticulin, and PKC, which transduce the biological responses triggered by Ca^2+^. Furthermore, MEL binds to the quinone reductase-2 (the MT3 receptor) in the cytosol and the orphan retinoid receptors in the nucleus [61,62].

The participation of MEL receptors in all stages of neurodevelopment has been evidenced by the administration of luzindole, a non-selective antagonist of these receptors [40,49,63,64], and also by using the pertussis toxin that uncouples adenylate cyclase, which inhibits the downstream signaling of MT_1_ and MT_2_ receptors [62]. The investigations of Sotthibundhu [41] and Tocharus [65] support these findings.

In addition, in the first stage of neurodevelopment, that is, in proliferation and cell survival, MEL receptors signaling downstream activate fibrosarcoma kinases, mitogen-activated protein kinase, and extracellular signal-regulated kinase 1 and 2 (Raf/MEK/ERK1/2). These kinases stimulate transcription factors that lead to gene expression [65]. Remarkably, similarly to MEL, this downstream pathway is activated by BDNF [29,56].

During neurodevelopment, there is a higher energy demand and an increased oxygen consumption with a higher ATP/ADP turnover rate. MEL enhances mitochondrial metabolism, increasing the expression of mitochondrial DNA. Therefore, the indolamine participates in the formation of mitochondrial mass and the development of oxidative phosphorylation complexes which, ultimately, confer energy to cells [66]. Furthermore, MEL as a free radical scavenger protects immature neurons by eliminating reactive oxygen species generated during this process [67]. The voltage-dependent L-type membrane channels are further MEL signaling pathways involved in neural differentiation. This molecule activates protein kinases A/B, which causes a transitory increase in the intracellular Ca^2+^ concentration and thus activates CaM [68]. In this regard, dendrite formation and arborization (which is a crucial step for the establishment of neuronal connections in neurodevelopment) are stimulated by MEL through autophosphorylation of calcium calmodulin kinase II (CAMKII), activation of PKC, and the phosphorylation of ERK 1/2. The involvement of these pathways is evident because dendrite formation stimulated by MEL is inhibited by KN-26, an antagonist of CaMKII, and by bisindolylmaleimide, an inhibitor of PKC [49]. Thus, data suggest that PKC acts upstream of CAMKII and downstream via MT_1_ and MT_2_ receptors [49,68].

Studies with cellular lines, such as human olfactory neuronal precursors, verified MEL effects in pluripotential cells [36,63,69,70]. These results concur with the effects observed in PC12 cells, mouse fibroblasts, and cells isolated from amniotic fluid [69]. Not only do NTFs elicit similar responses as MEL; MEL concentration determines the differentiation of pluripotential cells at a specific lineage. The indolamine drives differentiation to the neuronal lineage at 10^−7^ M (physiological concentrations), or to oligodendrocytes at 5 × 10^−6^ M. Hence, MEL concentration is critical to governing cell differentiation at specific lineages [71]. The mechanisms involved in MEL’s recruitment of cell linage are the activation of the tyrosine kinase signaling cascade, the ERK-mediated cell proliferation pathway, and the activation of intracellular Ca^2+^ and CAMKII [49]. In concordance with this evidence, we recently found that MEL, by binding to CaM in presence of Ca^2+^ and in an aqueous microenvironment, makes this protein adopt a specific structural conformation able to increase the activity of CAMKII, as shown in Figure 4 [72].

Finally, it is worth mentioning that the Jun-kinase pathway is one of the most important pathways activated by NTFs. The signaling of NFTs involves protein 53 (p53) activation and apoptosis. Among the various targets of p53 is the proapoptotic gene Bax [73]. Activation of one of the NTFs receptor p75NTR can also control the activity of Rho GTPase proteins, resulting in the inhibition of axonal growth and thus leading to selective pruning in neurodevelopment. In contrast, MEL modulates the apoptotic process via COX-2, p300, and the nuclear factor kappa β (NF-κβ) signaling, and, in addition, it suppresses p300 histone acetyltransferase (HAT) activity and p300-mediated NF-κβ acetylation [74]. The information recapitulated shows that the effects of MEL are comparable to other NTFs’ functions, such as the proliferation of neuronal precursors, their survival, and their consequent differentiation between neurons or glial cells. Both Mel and NTFs share IP3 and Ca^2+^-CaM signaling pathways.

## 5. Therapeutic Implications

As we have described, the regulation of neurodevelopment through mechanisms elicited by NTFs allows to claiming that the use of these molecules could be beneficial in the treatment of neuropsychiatric diseases. Moreover, neurogenesis studies have been a valuable tool for the development of new therapeutical drugs for neurodegenerative and affective disorders. In this context, it is important to mention that most antidepressants cause the release of NTFs that in turn stimulate neurogenesis in the hippocampus [75], synaptogenesis in the prefrontal cortex [76], and the production of neurogenic transcription factors [26,27,29,58,77,78,79]. One limitation in the treatment of affective disorders with NTFs is that they have poor pharmacokinetics and bioavailability, in addition to their inability to cross the blood-brain barrier [27]. Hence, several efforts have been directed to develop new molecules with other physicochemical features. In this regard, MEL as an amphiphilic molecule can cross the blood-brain barrier and its endogenous effects can be potentialized by exogen administration.

Evidence accumulated in the last decade shows that MEL antidepressant-like effects in rodents are associated with increased neurogenesis in the dentate gyrus of the hippocampus [11,80,81,82]. Administration of MEL reduced the immobility time in the forced swimming test (FST) and in the tail suspension test (TST) paradigms, providing evidence for the antidepressant effects of MEL [83,84]. In this respect, one must add that the effect of MEL was potentiated depending on the time of administration [82].

The neurogenesis stimulation by MEL is documented in rodents that were submitted to behavioral paradigms. For example, a study with BalbC mice reports that intraperitoneal administration of MEL for 14 days potentiates the effect of citalopram, an SSRIs antidepressant, and stimulates neurogenesis in the hippocampal dentate gyrus [85]. Moreover, we recently showed that triple administration of non-effective doses of MEL combined with low doses of ketamine elicits antidepressant-like effects in Swiss Webster mice and also increases neurogenesis in the dentate gyrus hippocampus [43]. Noteworthy is that this combination increased neurogenesis in a clone derived from human olfactory neural precursors similar to FGF, EGF, and BDNF neurotrophic factors [44]. In addition to being effective as an antidepressant in mouse models, MEL reduces the levels of proinflammatory interleukins and TNF-*α* release, likewise, MEL reduces oxidative stress and increases BDNF expression [83]. MEL is also useful in the treatment of other neuropathic or neuropsychiatric diseases, such as fibromyalgia, a chronic muscle-skeletal disorder characterized by generalized muscular pain and chronic fatigue, sleep disruption, depression, and anxiety. In a clinical study, MEL was found to decrease anxiety, pain, stiffness, and depressive symptoms similarly to fluoxetine in a concentration-depended manner [86]. The anti-inflammatory and antidepressant effects of MEL are exerted by the scavenger properties of MEL [87,88] and through its receptors [89,90]. For this reason, MEL could prove useful for treating inflammatory and pain-related diseases [90].

In neurodegenerative diseases, such as AD, Huntington’s, or Parkinson’s, MEL can increase cell survival in specific brain regions, such as the cerebral cortex [91,92]. To support this claim, it is important to mention that AD, which is a complex neurodegenerative disorder, is characterized by oxidative stress and developed by constant overproduction of free radicals coming from different pathways in areas where β-amyloid forms aggregate and trigger an imbalance of state homeostatic redox. Thus, MEL as a scavenger of free radicals modulates the neuroinflammatory response, diminishes the oxidative stress, and directly interacts with β-amyloid, preventing its aggregation [93]. Additionally, therapeutic MEL effects in AD are also possible thanks to the activation of the MT_1_ receptor and downstream pathways. Together they promote transcription proteins; for instance, the expression of the cAMP-response element binding protein gene (*Creb1*) and *Bdnf* lead to an increase in neurotrophic factors in the hippocampus and reduce memory impairment [80]. Learning and memory impairment in a model of AD in mice can be improved by MEL. Administered as a prophylactic, Mel can up-regulate CREB/BDNF signaling and cholinergic transmission in the prefrontal cortex [81].

Furthermore, MEL reduces age-related cognitive impairment [94]. Factors such as weight gain [95], constant exposure to stress [96], or even the use of medications for the treatment of chronic diseases [97,98] are risk factors in the development and progression of cognitive loss. For example, in a model of induced obesity in C57BL6 mice, cognitive functions were, on the one hand, evaluated with behavioral tests, and, on the other, the inflammatory cytokines associated with cognitive impairment and the BDNF levels as neurogenesis markers were also determined. In comparison to the control group, mice treated with MEL for eight weeks showed a reduction in inflammatory cytokines as well as an increase in BDNF [95]. Moreover, MEL can potentiate its effects when accompanied by physical activity [99]. The chronic administration of MEL also diminished proinflammatory cytokines levels in aged mice, demonstrating that its use may prevent memory impairment in aging [94]. In addition, MEL found in different food sources including walnuts, may be part of the mechanism involved in lowering inflammation and further preventing some age-related diseases, thus should be useful in the intake of these seeds [100].

Additionally, treatment with increasing doses of MEL is effective for improving the failed cognition induced by chronic stress exposure in rats, enhancing mood state, neurogenesis, and synaptogenesis [96]. The prolonged administration of MEL as a prophylactic treatment has been proven to enhance neurogenesis. For instance, in a chronic stress model of sleep deprivation, mice treated with the indolamine showed that in comparison to the non-treated groups, MEL promotes antiapoptotic and antioxidant molecules [101]. Hence, this molecule could alleviate neuroinflammation, cognitive, and mood impairments.

Indeed, the characteristics of MEL related to its structure as an antioxidant, the nature of its receptors, and its signaling pathways confer neuroprotective effects, for example, MEL increased the effect of antioxidant enzymes, maintained the balance of free radical production, and increased the production of BDNF in rodent models subjected to a neurotoxic administration scheme with methotrexate, which is used for the treatment of cancer [97], or valproate which is used in the treatment of epilepsy [98].

Furthermore, in neurodegenerative diseases like Multiple Sclerosis, MEL treatment increases neurogenesis and transcriptional markers that lead to neuron survival in the mouse hippocampus [102]. In a neurodegenerative animal model, the transplant of neurons cultured with MEL developed and increased proliferation and differentiation of neurons at the transplanted brain region and shows a clear contrast with the transplant without MEL treatment [66]. Despite the proposed therapeutic uses for MEL (Table 1), it is crucial to further explore its mechanisms and whether combining MEL with other substances could potentiate its effects. For example, the spinal or supraspinal MEL system might be involved in the modulation of neuropathic pain because MEL intrathecal and oral treatment has been shown to relieve the pain and deceased tactile allodynia induced by spinal denervation in rats; the antiallodynic MEL effect was here mediated by the MT_2_ extrapineal receptor and opioid receptors, possibly by increasing the β-endorphin release [103]. However, specific research is required in this field to indicate whether this type of therapy does not affect other brain regions and whether neural connections develop efficiently in the long term and without affecting the behavior of individuals.

## 6. Final Considerations: Is Melatonin a Neurotrophic Factor?

In this review, we collected evidence indicating that MEL is a factor that modulates neuronal survival, proliferation, differentiation, apoptosis, and the structural polarization of neurons (Table 1). Remarkably, NTFs and MEL act similarly although they are chemically distinct. The former are peptides in nature while the latter is an indolamine with a low molecular mass. In addition, MEL is endogenously produced like NTFs and its actions can be mediated by specific receptors that activate signaling pathways promoting these processes. In addition, and similarly to NTFs, MEL stimulates neurogenesis in organisms at the early stages of their development and in the adult brain, promoting dendritogenesis and the extension of dendritic trees, particularly in the hippocampus.

Despite considerable efforts made in this field by specialists, MEL’s mechanisms of action have not been thoroughly understood and more knowledge gaps arise when it is used in combination with other substances. For this reason, it is necessary to conduct more research at the molecular level to determine, firstly, whether these types of interactions exist and, secondly, to understand these mechanisms in depth together with the signaling pathways that are activated to conduct their functions.

## 7. Method

This review was conducted through a broad search of information using books and the PubMed and ScienceDirect databases. To detect original and review articles, the following search terms were used: MEL AND neurotrophic; MEL AND central nervous system; MEL AND trophic factor; MEL AND neurogenesis; MEL AND neuronal precursor cultures (NSCs); MEL AND neural differentiation. Titles and abstracts were read and inclusion and exclusion criteria were applied.

The inclusion criteria considered the following (1) articles from original and review journals, (2) books or book chapters with ISSN registration, (3) documents in English, (4) articles published in the last ten years, except for seminal or high-impact articles in the area, for which the year of publication was not considered. The following exclusion criteria were applied: (1) articles that did not present the search terms in the title and abstract; (2) repeated articles, (3) journal articles without an impact factor. One author (AMR) performed data extraction which was confirmed by others (GBK, RER, and GO). Search results are displayed in the description and in tables and figures.

## Figures and Tables

**Figure 1 molecules-27-07742-f001:**
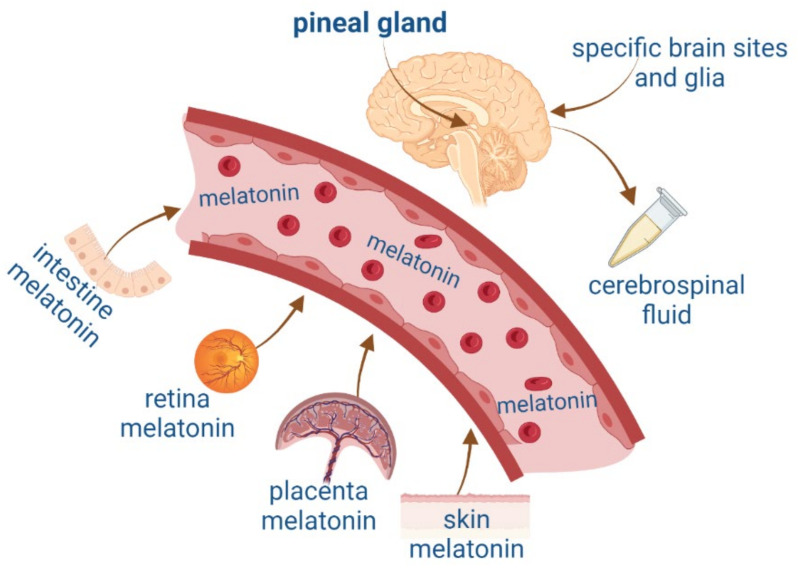
Melatonin synthesis in the pineal gland and extrapineal sites such as the intestine, retina, placenta, and skin and their blood circulation. Figure created by Biorender.com.

**Figure 2 molecules-27-07742-f002:**
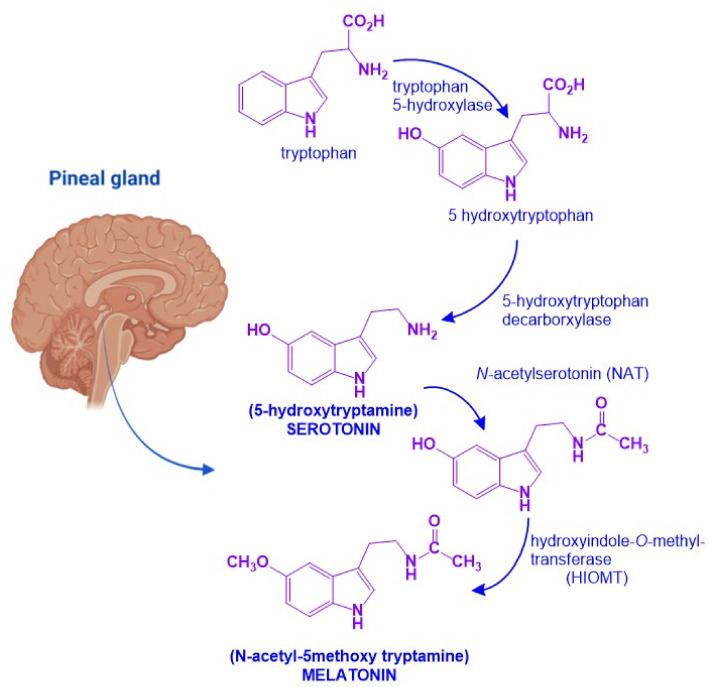
Melatonin biosynthetic pathway. Melatonin biosynthesis in the pineal gland starts from tryptophan and involves four sequential enzymatic steps to render 5-hydroxytryptophan, *N*-acetylserotonin, 5-hydroxytryptamine (serotonin), and *N*-acetyltryptamine (melatonin). Figure created with Biorender.com software.

**Figure 3 molecules-27-07742-f003:**
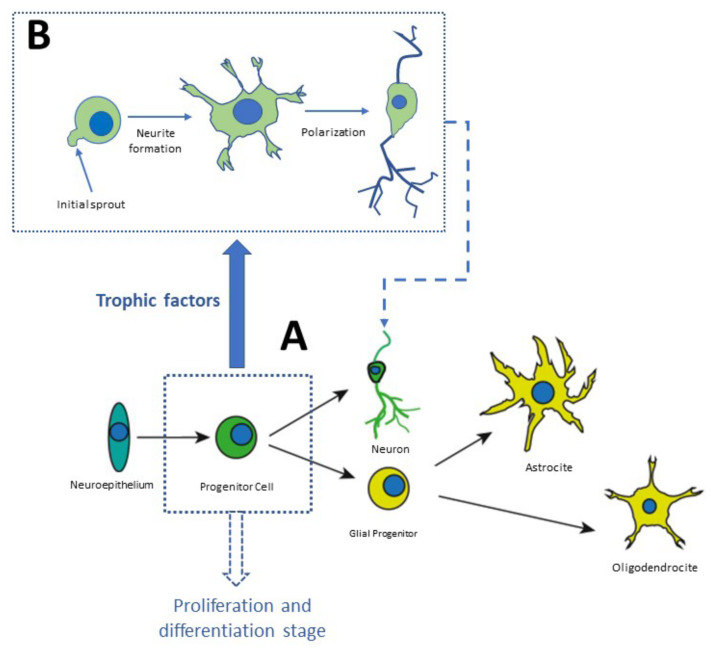
Stages of neurodevelopment in vitro. Stem cells derived from the neuronal niches located at specific regions in the brain or in the olfactory neuroepithelium can be plated in Petri dishes. In culture, they proliferate and amplify the population of progenitor cells which differentiate into neurons or glial cells, as shown in panel (**A**). Panel (**B**) shows distinct phases of differentiation [35,36].

**Figure 4 molecules-27-07742-f004:**
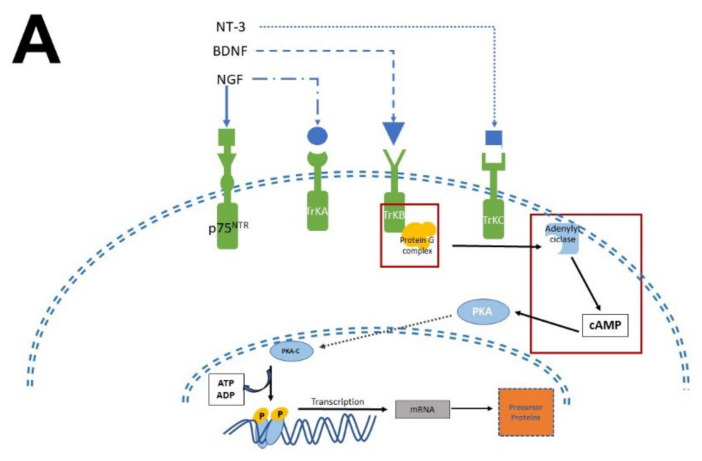
Simplified scheme of neurotrophic factors and melatonin signaling pathways. Panel (**A**) shows the interaction of neurotrophins (NT-3, BDNF, NGF) with their receptors (p75NTR, TrkA, TrkB, TrkC), the activation of G proteins, which in turn activate protein kinase A (PKA) and protein kinase C (PKC) to generate transcription factors and precursor proteins that promote neuronal survival. Panel (**B**) display melatonin (MEL) signaling pathways that promote cell proliferation through binding to melatonin receptors (MT_1_, MT_2_), which activate the protein G complex and fibrosarcoma kinases (Raf), mitogen-activated protein kinase (MEK) and extracellular signal-regulated kinase (Erk). MEL can also cross plasmatic membranes and bind to calmodulin (CaM); the complex CaM/MEL binds to the enzyme calmodulin kinase II and enhance its activity to stimulate dendritic development and growth. Red squares in both panels indicate the common features in these pathways.

**Table 1 molecules-27-07742-t001:** MEL effects that support its neurotrophic actions.

MEL Neurotrophic-like Characteristics	References
MEL has membrane receptors coupled to G protein	[54,61]
MEL promotes neuronal differentiation, proliferation, and neuronal survival in newly formed neurons	[39,40,41,63,64,65,66,69,71,104,105]
MEL participates in mature brain neuroplasticity and neurodevelopment by stimulating second messenger cascades (Ca^2+^, CaMKII, Trk receptors)	[63,65,69,80,81,106]
Exerts antidepressant and anxiolytic-like effects associated to increased neurogenesis	[44,45,50,64,80,81,85]
Adjuvant in antidepressant treatments due to its synergistic effect with other molecules administered as antidepressant such as SSRIs (fluoxetine, citalopram) or ketamine	[43,44,85,86]
MEL can be used as neuroprotector in diseases whose treatments cause toxicity	[97,98]
MEL has anti-inflammatory properties and restorative effects in toxin-induced rodent models of persistent/chronic and neuropathic pain and spinal cord injury	[66,87,88,89,90,94,95,96,103]
MEL protects and maintains immune cells in non-neuronal places, like NTFs does	[14,19]
MEL can change the reorganization of the cytoskeleton	[46,47]
Promotes neuron’s maturation, neuritogenesis, dendritic growth, and axogenesis. MEL also increases the complexity of the dendrite trees in crucial brain regions affected in neurodegenerative diseases.	[37,45,46,47,48,49,50,51]
Free radical scavenging properties of MEL equip it with antioxidant and neuroprotective effects. MEL modulates the production of other antioxidant molecules	[17,19,66,87,88]

## Data Availability

Not applicable.

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
