# Peer review of "Melatonin: A Neurotrophic Factor?"

_molecules, 2022, doi:10.3390/molecules27227742_

Round 1

Reviewer 1 Report

There are minor issues with the grammar:  For example, Line 80-83 should be re-written: For example, regarding the synthesis of melatonin by the retina, the most probable current hypothesis would be that of a local action of melatonin that would regulate retinal photosensitivity and that it would be rapidly catabolized in the same place from which the feeble influence on plasma melatonin concentration [9]. There are several other confusing sentences.

Generally, the authors were not sufficiently critical of the previous work.  They just presented it. They cited the substantial previous work suggesting that melatonin has pleiotropic effects (e.g., refs 16-20), so this part of their review is not novel.  In terms of their argument, that melatonin is also a neurotrophic factor, they cite many of the published articles from the past decade or so, but not critically.  For example, many of the studies on melatonin neurotrophic effects were conducted on strains of mice lacking melatonin or melatonin receptors--this should be mentioned.

Author Response

We are very grateful for the time taken to review the manuscript and all the comments made. We feel that this new version has certainly improved with your feedback. 
Please, find attached our point-by-point responses.

Reviewer 2 Report

This paper has an interesting and challenging topic. The authors highlighted only part of this research field; the manuscript needs to be better structured, tables improved, English revised, citation rechecked (there are authors in Table 3 not mentioned in References, e.g. Arendt 2004), references updated (some are too old, many new articles are not cited and discussed).

At a simple check in PubMed with the same search terms used by the authors revealed many titles with very interesting ideas that should have been mentioned in such a review. Following are some of these articles:

Melatonin significantly attenuated the levels of integrin, glial fibrillary acidic protein, interleukin-1 beta, interleukin-6, tumor necrosis factor alpha, and phosphor-nuclear factor kappa B in aged mouse brains (doi: 10.17179/excli2017-654).

Melatonin treatment significantly upregulated synaptic plasticity and an immature neuron marker through enhancing brain derived neurotrophic factor (doi: 10.1155/2022/1596362).

Melatonin treatment exerted neuroprotective effects against LPS-induced depressive-like behavior which may be related to reduction of TNF-α release, oxidative stress and modulation of BDNF expression (doi: 10.1016/j.physbeh.2018.02.034).

Melatonin induces CREB signaling pathways associated with long-term memory processing in vitro (doi: 10.3390/molecules23040737).

Melatonin ameliorated Creb1 gene expression and significantly increased Bdnf gene expression in the hippocampus of Alzheimer's disease model mice compared with the Alzheimer's disease group (doi: 10.2147/DNND.S291172).

Melatonin enhanced the cholinergic system and BDNF and CREB signaling pathways in the prefrontal cortex in an Alzheimer's disease mouse model (doi: 10.1016/j.bbr.2020.113100).

Melatonin was able to counteract neurotoxic effects (doi: 10.1155/2021/9997582)

Melatonin treatment effectively improved all cognitive impairments and reduced TNF-α, IL-17A and elevated BDNF levels in the hippocampus of obese mice (doi: 10.1016/j.physbeh.2022.113919).

Melatonin treatment ameliorated hippocampal neurogenesis in the dentate gyrus by improving BDNF levels (doi: 10.1002/brb3.1388).

Melatonin activated the BDNF-ERK-CREB signaling pathway, enhanced neurogenesis and diminished synapse loss in the hippocampus, and improved a more complex hippocampus-dependent cognitive function (doi: 10.1016/j.redox.2021.101973).

Melatonin increased BDNF and improved dysregulations of cytokines in the cerebellum (doi: 10.1002/npr2.12125).

I advise the improvement of the manuscript and resubmission.

Author Response

We are very grateful for the time taken to review the manuscript and all the comments made. We feel that this new version has certainly improved with your feedback.
Please see the attached document with our responses.

Reviewer 3 Report

The authors reviewed the potential neurotropic role of melatonin based on recent and less recent evidence of this pleiotropic neurohormone. My major recommendation is to be more specific in the use of the ref, since in several part of the ms they lack. Also, sometimes the table do not add any major info to the ms (particularly table 2 should be re-thought, since its role is not clear in the entire ms).

Following my comments:

69: please use the correct nomenclature “G-alpha-stimulatory, Gαs”.

72: Based on the current and solid literature, MLT's secretion increases soon after the onset of darkness, peaks in the middle of the night, between 2 and 4 a.m., and falls during the second half of the night. (Arendt, 2005) Please describe accordingly, or cite different literature discussing potential discrepancies.

139: add a citation. Ex: Khuen Yen Ng, 2017 and add MT2 expression in the brain too

140: “PKA inhibition concentrations”. It is not clear. PKA is inhibited due to decreased cAMP. Please rephrase.

139-144. Please cite relevant literature about MT1 and MT2 signaling, e.g. Dubocovich, 2010, 10.1124/pr.110.002832

145-148. please expand this part citing the relevant literature.

Table 1: please cite the literature reporting MLT affinity for the receptor. E.g. Dubocovich ML. (1985) and Audinot et al 2003. Please indicate affinity as pKi as it is more precise.

152: “3.2 Melatonin: a pleiotropic factor”. MLT’s effect as radical scavenger has been linked to MLT's antinflammatory properties (for a review see Posa et al. 2018 https://doi.org/10.2174/0929867324666170209104926). Starting from this evidence, please add a short paragraph mentioning the antinflammatory effect of MLT in acute and chronic condition as part of its pleiotropic effect.

155, 160, 181, 184, 190, 228: in all these statements, authors must cite the relevant literature.

207: “small quantities forty-four of one or more”. This sentence is not clear. Please rephrase it.

241: remove “and”

277: remove “ : “

279-282: authors should expand more this part about dendritogenesis, axogenesis and synaptogenesis.

289: pertussis

364: RTKs-GPCRs transactivation is a well-known phenomenon (for exhaustive reviews see https://doi.org/10.1016/j.tips.2007.09.007 and 10.1016/j.cell.2010.06.011). Is there any evidence of MT1 and /or MT2 cross-talking with FGFR, EGFR/ErbB-1 or TrkB ? How this potential transactivation can modulate neurogenesis. Discuss it with a particular focus on neuropsychiatric and neurodegenerative diseases.

Author Response

We are very grateful for the time taken to review the manuscript and all the comments made. We feel that this new version has certainly improved with your feedback.

Please, find attached our document wit point-by-point responses.

Round 2

Reviewer 2 Report

The paper has improved. However, all the authors have to read and correct the document. Please address the following questions and suggestions:

English should be revised throughout!

Acronyms/Abbreviations should be defined the first time they appear in each of three sections: the abstract; the main text; the first figure or table (when defined for the first time, the acronym/abbreviation should be added in parentheses); then, acronyms should be used.

All Figures and Tables should be inserted into the main text close to their first citation.

References should be revised throughout and the same form used! There are 91 references in the text mentioned by numbers, and many others by names! In the Reference section, there are only 67!!

Line 51: “... neurotrophic factors (NTFs) do [4,5].

Line 52: “In this regard, NTFs, play ...

Line 58: “NTFs are also required ...

Line 60: “lung components

Lines 143, 177, 180, 345: “NTFs.”

Lines 208-225: this citation can be added in this paragraph - “melatonin not only counteracted and protected against oxidation and apoptosis, but also prevented and reversed the dendritic arbor retraction” ( doi: 10.3390/molecules25235508).

Lines 213-215: please revise sentence

Line 234: “Also, an increase in the volume of CA3 region was observed in this work [49].

Line 237: “suggests”

Line 252: “There are...”

Lines 253-254: Trk was defined previously

Line 271: MT1 and MT2 shared...

Lines 273-274: “they activate”?

Lines 284-285: revise sentence

Line 292:  This can be added here - besides antioxidant activity, melatonin revealed anti-inflammatory properties” (doi: 10.3390/antiox11071412)

Line 301: “data suggest...”

Line 303: “It is worth mentioning that...

Lines 318-319: revise sentence

Line 325: NFκβ”?

Line 327: “factors” can be deleted

Line 332: is Figure 4. included in text?

Line 387: “Bdnf”?

Line 446: revise sentence

Author Response

Point 1: The paper has improved. However, all the authors have to read and correct the document. Please address the following questions and suggestions:

English should be revised throughout!

Response 1: The manuscript has been reviewed and edited by an expert in English language; We consider that the paper certanly was improved.

We corrected each of your suggestions. we made changes throughout the manuscrit. However,  the marks of the corrections made the manuscript reading very difficult. .

We greatly appreciate major your time presented to this paper. Your comments have contributed  to improve our work.

Point 2: Acronyms/Abbreviations should be defined the first time they appear in each of three sections: the abstract; the main text; the first figure or table (when defined for the first time, the acronym/abbreviation should be added in parentheses); then, acronyms should be used.

Response 2: The abbreviations have been corrected and properly used.

Point 3: All Figures and Tables should be inserted into the main text close to their first citation.

References should be revised throughout and the same form used! There are 91 references in the text mentioned by numbers, and many others by names! In the Reference section, there are only 67!!

Response 3: The references were revised throughout the manuscript and all were incorporated in the references section.

Point 4: Line 51: “...neurotrophic factors (NTFs) do[4,5].”

Response 4: This was corrected

Point 5:Line 52: “In this regard, NTFs, play ...”

Response 5: This was corrected

Point 6: Line 58: “NTFs are also required ...”

Response 6: This  was modified and corrected

Point 7: Line 60: “lung components”

Response 7: This was corrected

Point 8:Lines 143, 177, 180, 345: “NTFs.”

Response 8: T his was corrected

Point 9: Lines 208-225: this citation can be added in this paragraph - “melatonin not only counteracted and protected against oxidation and apoptosis, but also prevented and reversed the dendritic arbor retraction” ( doi: 10.3390/molecules25235508).

Response 9:  We agree with the referee’s suggestion. The citation was included the manuscript

Point 10: Lines 213-215: please revise sentence

Response 10: The sentence was modified

Point 11: Line 234: “Also, an increase in the volume of CA3 region was observed in this work [49].” Response 11:  The sentence was changed

Point 12: Line 237: “suggests”

Response 12: The word was corrected

Point 13: Line 252: “There are...”

Response 13: These  words were changed

Point 14: Lines 253-254: Trk was defined previously

Response 14: We used only the abbreviation for Trk

Point 15: Line 271: “MT1 and MT2 shared...”

Response 15: We agree with the observation. We changed this line; first, we defined melatonin receptors, and then, we used only the abbreviation.

Point 16: Lines 273-274: “they activate”?

Response 16:

We changed the wordss and improved for a best understanding

Point 17: Lines 284-285: revise sentence

Response 17: The sentence was changed and improved for the best understanding

Point 18: Line 292:  This can be added here - “besides antioxidant activity, melatonin revealed anti-inflammatory properties” (doi: 10.3390/antiox11071412)

Response 18:  This reference was considered, and incorporated into the manuscript

Samah Labban and others, ‘Melatonin Improves Short-Term Spatial Memory in a Mouse Model of Alzheimer’s Disease’, Degenerative Neurological and Neuromuscular Disease, 11 (2021), 15–27 <https://doi.org/10.2147/DNND.S291172>.

.Point 19: Line 301: “data suggest...”

Response 19: The words were corrected

Point 20: Line 303: “It is worth mentioning that...”

Response 20: The words were corrected

 Point 21: Lines 318-319: revise sentence

Response 21: The sentence was corrected

Point 22: Line 325: “NFκβ”?

Response 22: This was revised and corrected

Point 23: Line 327: “factors” can be deleted

Response 23: We agree with the observation anddeleted the word “factors”

Point 24: Line 332: is Figure 4. included in text?

Response 24: We agree and apologize for not including the title “Figure 4” in the text. We have already revised and included in this version.

Point 25: Line 387: “Bdnf”?

Response 25: We revised line 397 on the manuscript. The notation is correct. When it is named as a peptide Brain-derived Neurotrophic Factor, its abbreviation is written BDNF; when it is mentioned as a transcription factor or gene it is indicated with italics: Bdnf. Also in the reference taken, the authors wrote it in italics. We kindly request the referee look into the citation in the footnote [1]

Point 26: Line 446: revise sentence

Response 26: We revised and changed from “The inclusion of documents considered the following criteria” to: The inclusion criteria considered the following”

Reviewer 3 Report

The manuscript has been fully re-written, with major improvements.

My major comments focus on 2 points:

- The part concerning the antiinflammatory properties of MLT should be expanded. Since this is a comprehensive review, it should be at least mentioned the antinflammatory proeprties of MTL . E.g. :

Costantino, et at al.  Eur. J. Pharmacol., 1998, 363(1), 57-63. ; Bilici, et al, 2002, 46(2), 133-139. ; Cuzzocrea, et al,  J. Pineal Res., 1997, 23(2), 106-116.; Posa et al.  J. Pineal Res., 2020 69 (3), e12671; Ambriz-Tututi, et. al. Pharmacol. Biochem. Behav., 2011, 98(3), 417-424.

For an exaustive review, see Posa et al. 2018, Current medicinal chemistry 25 (32), 3866-3882.

- While the main body of the manuscript indicates 93 references, the "References" only 67 are listed. Please fix it.

Minor comments:

- 371-372.  Hussain, et al. Pineal Res., 2011, 50(3), 267-271 found that MLT decreases anxiety, pain, stiffness, and depressive symptoms compared to fluoxetine or MLT alone in patient with fibromyalgia. Please discuss it.

- Table 1 should be rethought and improved in its content to make it more exhaustive and rich in content

Author Response

The manuscript has been fully re-written, with major improvements.

My major comments focus on 2 points:

Point 1: The part concerning the antiinflammatory properties of MLT should be expanded. Since this is a comprehensive review, it should be at least mentioned the antinflammatory proeprties of MTL . E.g. :

Costantino, et at al.  Eur. J. Pharmacol., 1998, 363(1), 57-63. ; Bilici, et al, 2002, 46(2), 133-139. ; Cuzzocrea, et al,  J. Pineal Res., 1997, 23(2), 106-116.; Posa et al.  J. Pineal Res., 2020 69 (3), e12671; Ambriz-Tututi, et. al. Pharmacol. Biochem. Behav., 2011, 98(3), 417-424.

For an exaustive review, see Posa et al. 2018, Current medicinal chemistry 25 (32), 3866-3882.

Response 1:  

We included and mentioned the listed citations and agree with the importane to remark on the MEL's antiinflammatory effects. However, in this review, we focus on the melatonin neurotrophic effects.

Point 2: While the main body of the manuscript indicates 93 references, the "References" only 67 are listed. Please fix it.

Response 2: As the reviewer kindly indicated, we agree and apologize for sending a previous version with missing references. In this final version, the references have been revised throughout the manuscript and incorporated in the references section.

Minor comments:

Point 3: 371-372.  Hussain, et al. Pineal Res., 2011, 50(3), 267-271 found that MLT decreases anxiety, pain, stiffness, and depressive symptoms compared to fluoxetine or MLT alone in patient with fibromyalgia. Please discuss it.

Response 3: We thank the referee for the suggestion, we included the afore mentioned reference in the new version along with inflammatory references.

Point 4: Table 1 should be rethought and improved in its content to make it more exhaustive and rich in content.

Response 4: We improved Table 1 by briefly mentioning the melatonin neurotrophic characteristics and incorporated the references in the molecules format.

We are very grateful for the time taken to review the manuscript and all the comments made in rounds 1 and 2, which undoubtedly improved our final version of the manuscript.

Round 3

Reviewer 2 Report

The manuscript has been improved considerably; would recommend it for publication after a few details have been attended to:

Line 199: “Alzheimer’s disease (AD)”

Line 311: the nuclear factor kappa B (NF-κB) - if it was not previously defined 

Line 330 (Therapeutic implications): the following is an interesting idea that can be added to this section - “Melatonin, found in different food sources including walnuts, may be part of the mechanism involved in lowering inflammation and further preventing some age-related diseases. (doi: 10.3390/antiox11071412)

Line 369: “Alzheimer’s” can be deleted - it was defined at line 199

Author Response

Thank you very much for the corrections. 

We attend all of them and certainly the paper was improved.

Reviewer 3 Report

Thanks you for this revised version.

Minor comment:
255-256 : "MT1 and MT2 receptors are coupled to G proteins, which activate either phospolipase C or adenylate cyclase and generate IP3 and cAMP, respectively[54,61]". This sentence is a bit confused. GPCRs can be coupled to different kind of Galpha proteins, some are stimulatory (Gs) some inhibitory (Gi). MT1-2 can be coupled to Gi which in turn inhibits the cAMP production or to a Gq which activate beta-type PLC-β leading to the production of PIP2, IP3 and DAG (For a review see https://doi.org/10.2218/gtopdb/F39/2021.3). Please rephrase it based on this information.

Author Response

Thank you very much for the suggestions.

We corrected the phrase and review the nomenclature and corrected through out the manuscript.